# The Cost-Effectiveness of Organized National Colorectal Cancer Screening Program in Croatia

**DOI:** 10.3390/cancers18010150

**Published:** 2025-12-31

**Authors:** August Cesarec, Nataša Antoljak, Ivana Brkić Biloš, Mario Šekerija, Maja Vajagić, Neven Ljubičić

**Affiliations:** 1Faculty of Economics and Business, University of Zagreb, 10000 Zagreb, Croatia; august.cesarec@gmail.com; 2Croatian Institute of Public Health, 10000 Zagreb, Croatiamaja.vajagic@hzjz.hr (M.V.); 3School of Medicine, University of Zagreb, 10000 Zagreb, Croatia; 4Sestre Milosrdnice University Hospital Center, 10000 Zagreb, Croatia; 5School of Dental Medicine, University of Zagreb, 10000 Zagreb, Croatia

**Keywords:** colorectal cancer, screening, gFOBT, fecal immunochemical test, cost-effectiveness, cost of illness

## Abstract

The organized National Colorectal Cancer Screening Program in Croatia was introduced by the government in 2007. Every two years, approximately 1,300,000 individuals aged 50–74 years are invited by mail to participate in guaiac fecal occult blood test (gFOBT) screening, and positive patients are referred for colonoscopy. Previous research has shown that colorectal cancer screening programs not only reduce disease-specific mortality but are also generally cost-effective. This study evaluates the costs of colorectal cancer screening in Croatia and compares the cost effectiveness of three screening strategies in the population aged 50 to 74 years: no screening, biennial gFOBT, and biennial fecal immunochemical testing (FIT). The results indicate that the introduction of FIT screening represents a cost-effective strategy for colorectal cancer screening in Croatia.

## 1. Introduction

Colorectal cancer (CRC) was the third most commonly diagnosed cancer (9.6% of all cases; following lung and breast cancer) and second by the number of deaths (9.3%; after lung cancer) worldwide in 2022 [1]. In the Republic of Croatia, it was the most frequently diagnosed cancer and second by mortality (after lung cancer) among all cancers [2]. According to estimates from the European Cancer Information System for 2022, Croatia ranks four in the incidence and first in mortality of colorectal cancer among the 27 European Union (EU) member states [3]. According to the CONCORD-3 study, Croatia ranks at the lower end among EU countries in 5-year net survival (51.2% for colon cancer and 48.2% for rectal cancer for patients diagnosed in period 2010–2014) [3,4]. Beyond its health related impact, CRC imposes a significant economic burden, with estimated annual costs of USD 24 billion in the USA and EUR 19 billion in Europe [5,6].

Among all cancer types, some malignancies, such as colorectal cancer, can be detected at an early stage through organized population-based screening programs. Available screening tests include the guaiac fecal occult blood test (gFOBT), fecal immunochemical test (FIT), colonoscopy, sigmoidoscopy, and other emerging tests (blood-based DNA test, double-contrast barium enema, etc.) [7]. Screening could potentially improve clinical outcomes by enabling earlier cancer diagnosis and treatment. For example, in the Netherlands, the implementation of a national FIT-based screening program increased the proportion of colorectal cancer cases diagnosed at Stage I from 15% prior to screening to 48% after screening [8].

In the European Union, the guaiac fecal occult blood test (gFOBT) and the fecal immunochemical test (FIT) are the most widely used screening methods. While gFOBT detects blood in the stool using guaiac and requires dietary restrictions, FIT uses antibodies to detect human hemoglobin and does not require dietary restrictions. Owing to its higher sensitivity and greater patient acceptability, FIT is considered the superior screening option [9].

By 2010, 20 member states of the European Union established a population-based screening program for colorectal cancer [10]. Unfortunately, coverage of the target population for colorectal cancer screening remains uneven, ranging from 20% to 80%. The participation rates are influenced by the type of screening test, with higher participation in the countries which used FIT screening test [11]. In November 2022, the Council of the European Union issued updated recommendations for a new EU approach on cancer screening, endorsing FIT as a preferred screening test for colorectal cancer screening [12].

Given the variety of available screening tests, numerous studies have assessed their relative cost-effectiveness solution. Based on available systematic reviews, the majority of studies conclude that screening interventions are cost-effective relative to no screening [13,14]. Several studies also compared different screening methods on a national level. A study for a Basque county in Spain concluded that the current screening program, i.e., biennial FIT, dominated no screening by increasing life expectancy while reducing costs. The cost-effectiveness of FIT versus g-FOBT was proven in the UK setting, yielding both cost savings and gains in quality-adjusted life years [15].

The aim of this study is to evaluate costs of colorectal cancer treatment and screening in Croatia and to compare three screening scenarios: no screening, biennial gFOBT test for individuals aged 50–74 years, and biennial FIT for the same age group. The study explores two research questions: (1) Are the costs of colorectal cancer treatment substantially higher at advanced cancer stages? and (2) Does the inclusion of FIT test have the potential to be cost-effective in the Croatian healthcare system? Our main hypothesis is that implementation of FIT test could improve treatment outcomes and be cost-efficient strategy in the Croatian setting.

## 2. Materials and Methods

### 2.1. Sources of Data

The data for this study were obtained from three primary sources: the Croatian National Cancer Registry, the Croatian National Colorectal Cancer Screening Program Registry and the Croatian Health Fund Claims Database.

First, the Croatian National Cancer Registry, maintained by the Croatian Institute of Public Health, has collected data about all newly diagnosed cancer cases in Croatia since 1962. Each registered case includes detailed information on such the date of diagnosis, disease stage, cancer topography and morphology, demographic patient characteristics (e.g., date of birth, sex, date of death,). The data sources for cancer registration are compiled from multiple sources, including hospital discharge notifications (“Onco type forms”), outpatient “Malignant neoplasm notification”, and copies of the histological/cytological findings [2].

Second, the National Colorectal Cancer Screening Program Registry collects relevant data for the organized population-based screening program in Croatia. Key indicators recorded in the registry include rate of compliance, the number of individuals with positive gFOBT, the number of patients invited to undergo colonoscopy, the number of colonoscopies performed, and the number of colorectal cancers and polyps detected [16].

Third, the Croatian Health Fund Claims Database provides detailed information on healthcare utilization and associated costs, derived from mandatory health insurance and contracts with healthcare providers. For the purpose of this analysis, relevant data were extracted on the inpatient (hospital) care, outpatient specialist care (specialist care in hospitals), primary care, prescription medications, orthopedic devices, and sick leave utilization.

### 2.2. Screening

In the Republic of Croatia, the National Colorectal Cancer Screening Program was established in 2007. The program targets population aged 50–74 years and is conducted every 2 years [17]. Screening is based on the gFOBT test, using the HemoGnost (Biognost) card test (detection limit: 0.252–0.348 mg hemoglobin/g of stool). Individuals with a positive gFOBT result are referred for diagnostic colonoscopy. Colonoscopies are performed in colonoscopy units (38 nationwide) by well-trained and experienced gastroenterologists.

Program performance is evaluated by an expert Committee for Program Coordination, which includes representatives from all medical specialties involved in the screening program. A web-based colorectal cancer screening registry enables the generation pre-defined reports and monitoring of key screening indicators. 

To participate in the screening program, each invited individual is required to submit stool samples on three test cards (each containing four windows) by prepaid mail to the designated institute for laboratory analysis. When used in population-based screening, the test does not require dietary restrictions, with the exception of avoiding high-dose vitamin C supplementation (>250 mg/day). To enhance participation and ensure correct test use, printed strip-form procedural instructions are provided on the back of the manufacturer’s instruction leaflet.

### 2.3. Croatian Healthcare System

In Croatia, healthcare insurance coverage is compulsory and every Croatian citizen should have a mandatory health insurance. The sole provider of this insurance scheme is the Croatian Health Insurance Fund, covering approximately 99% of the [18].

The Croatian Institute of Public Health, which coordinates the National Colorectal Cancer Screening Program, is funded by the Ministry of Health through the state budget in accordance with the annual program plan. Regional and local institutes of public health are financed by the Croatian Health Insurance Fund for their public health activities via payments to preventive public health teams. The Croatian Health Insurance Fund also reimburses preventive colonoscopies performed within the National Colorectal Cancer Screening Program based on special contracts with hospitals. Two reimbursement prices are defined, depending on whether the colonoscopy is performed with or without polypectomy.

### 2.4. Population

The study population consisted of newly diagnosed colorectal cancer cases in Croatia in 2014. Cases were retrieved from the Croatian National Cancer Registry. The following diagnoses were included according to the International Classification of Diseases: C18 (malignant neoplasm of colon), C19 (malignant neoplasm of rectosigmoid junction), C20 (malignant neoplasm of rectum), and C21 (malignant neoplasm of anus and anal canal).

According to registry data for 2014, a total of 3644 new colorectal cancer cases were diagnosed in Croatia. For contextual reference, the total population of Croatia was 3,871,833 in 2021. Three categories of patients were excluded from further analysis (Figure 1).

First, patients diagnosed through death certificate notifications were excluded (*n* = 221 patients, all patients have identical date of diagnosis and date of death). Second, patients without any recorded medical claims and healthcare costs from 2014 to 2019 were excluded (*n* = 16). Third, patients for whom the recorded date of diagnosis in the registry occurred after the date of death were excluded (*n* = 3). After applying these exclusion criteria, the final analysis comprised 3404 patients, of whom 2089 had the eligible age range for screening (50–74 years).

The mean age at colorectal cancer diagnosis was 68.9 years. There were more male patients in the study population than females (59.3% vs. 40.7%) and the mean age of diagnosis was similar between genders (68.2 for males and 69.8 for females). With respect to disease stage at diagnosis, 21.7% of patients were diagnosed at a localized stage, 43.3% at a regional stage, and 14.6% at a distant stage, while 20.4% had an unknown stage recorded in the registry. The relatively high proportion of cases with an unknown stage at diagnosis indicates a limitation in data completeness within the national cancer registry, as documented in its annual reports [2]. Determining the exact disease stage for these patients would require access to individual patient data and medical records. However, such access is difficult due to data sensitivity and the requirements of the European Union General Data Protection Regulation (GDPR). Consequently, cases with an unknown stage were treated as a separate (additional) disease stage in the costs and cost-effectiveness analyses.

### 2.5. Costs

Costs for each patient included in the analysis were derived from Croatian Health Insurance Fund Claims Database and covered five-year period following the date of diagnosis. As the Croatian National Cancer Registry records the disease stage only at the time of diagnosis and does not provide information on subsequent stage progression, the colorectal cancer-related costs were assigned according to the stage of diagnosis. For example, if a patient had a local stage at a diagnosis, all further healthcare costs were attributed to the localized disease category. The same rule was applied to all disease stages and patients. Regarding calculation of annual per patient costs, costs incurred in the second year were averaged among patients who survived the first year within each disease stage. Similarly, costs in the third year were averaged among patients who survived the second year within each disease stage. The same approach was applied for the fourth and fifth years. 

All healthcare claims in which one of the diagnoses C18–C21 was recorded as a primary or secondary diagnosis were included in the analysis. The total five-year costs for the 3404 patients amounted to EUR 44.6 million (Table 1). The largest proportion of costs was incurred in the first year following diagnosis (52.2%), with expenditures decreasing in each subsequent year. Regarding the type of healthcare, inpatient care accounted for the highest share of total costs (50.8%), followed by outpatient care (21.2%) and orthopedic and medical devices (20.2%). When compared with the total annual healthcare expenditures of the Croatian Healthcare Fund in 2014 and 2015 (approximately EUR 3.0 billion annually), the costs of treating newly diagnosed colorectal cancer patients in the first year after diagnosis represented approximately 0.8% of total expenditures, indicating a high burden on the healthcare budget.

Across colorectal cancer stages, the largest share of total costs was associated with regional stage (44.8%), followed by distant metastases (22.0%), unknown (17.2%), and local stage (16.0%). Estimates of per-patient costs indicate that the highest expenditures were associated with distant disease stage (cumulative five-year costs of EUR 39,802 per patient), followed by the regional stage (EUR 16,732 per patient) (Table 2). A comparison between costs for overall patient population and patients within the screening eligible age group shows higher average costs among screening eligible patients (average costs of EUR 19,533 vs. EUR 16,897). In both groups, patients diagnosed at a distant stage incurred more than 130% higher costs compared with those diagnosed at localized or regional stages. These findings emphasize the importance of earlier disease detection, not only from a clinical perspective but also in reducing economic costs.

Analysis of annual per-patient costs indicates a high economic burden of disease during the first two years since the diagnosis (Table 2). In the first year, depending on disease stage, costs accounted for approximately 29–45% of total five-year costs, while the first two years combined accounted for 58–67% of total costs. The proportion of costs incurred in the first year was the highest for the localized cancer stage and lowest for the distant stage, while in the second year it was highest for the distant stage and lowest for the localized stage. Costs for all cancer stages decreased in each subsequent year following diagnosis and were generally higher among patients within the screening (except for patients with an unknown stage in the third, fourth, and fifth years).

### 2.6. Model and Inputs

This cost-effectiveness analysis estimates costs, life-years gained, and the number of prevented colorectal cancer cases under three screening strategies: no screening, biennial gFOBT test for individuals aged 50–74 years, and biennial FIT test for individuals aged 50–74 years. The model adopts the perspective of the public healthcare budget in Croatia. The model projects health outcomes and associated costs over a five-year period for a cohort of 10,000 colorectal cancer free individuals aged 50 at baseline.

The predicted outcomes for five years include treatment costs, number of newly colorectal cancer patients (incidence), number of deaths from colorectal cancer, life-years gained (if patient with cancer is alive, it counts as 1 life-year gained), and costs or savings per life-year gained. Both costs and life-years gained were discounted at an annual rate of 3%, which is the most commonly applied discount rate in similar studies, as identified in two literature reviews [12,13].

It was assumed that patients may develop an adenoma or an advanced adenoma based on prevalence from the literature survey. A proportion of colorectal adenomas progress to advanced adenomas and a proportion of advanced adenomas progress to colorectal cancer. These transition probabilities, as well as an incidence of adenomas, were incorporated into the developed natural history model.

Patients were assumed to have undergone a screening in the first year according to observed participation rates for both the initial test and follow-up colonoscopy after a positive result. Participation rates were derived from data from the National Colorectal Cancer Screening Program and previous screening cycles in Croatia. Patients who participated in screening, tested positive, and underwent a colonoscopy could have adenomas and advanced adenomas detected and removed. The removal of these lesions reduces the future incidence of colorectal cancer, thereby improving survival and decreasing associated healthcare costs.

The five-year relative survival probabilities varied by cancer stage and type of detection (symptom detected vs. screening detected). Due to the unavailability of Croatian survival data stratified by detection mode, survival estimates were derived from Austrian data. Cancer stage distribution was based on Croatian colorectal cancer registry data for the period 2015–2019 (unknown stage 32%, localized 15%, regional 39%, and distant 14%). Relative survival probabilities for cases with an unknown stage were estimated as the average of localized, regional, and distant cancer stage survival probabilities. In the absence of national data for stage transition rates, it was assumed that patients remain in the same disease stage throughout the five-year model horizon. The model also used all-cause mortality estimates using Croatian life tables for the period 2010–2012 [19]. Screening test characteristics (sensitivity and specificity), as well as adenoma incidence and transition rates, were obtained from the literature search.

Costs per patients for colorectal cancer were calculated using insurance claims data from the Crotian Health Insurance Fund, as described earlier in this study (Table 2). Because the screening and model focuses on individuals aged 50–74 years, cost estimates specific to this age group were applied in the model. Regarding the costs of tests, they include the unit cost of the test and operational costs of the screening program based on the current gFOBT screening program. Costs of the screening program include postal services, education, quality control and marketing, public healthcare team activities, and printed materials. For the FIT-based screening, the costs of the screening program additionally include extra postal costs due to different packaging, acquisition of laboratory equipment, and additional laboratory personnel. Described screening program costs were calculated on a per-patient basis by dividing the total program cost by the number of individuals participating in the screening. All model inputs and sources are summarized in Table 3.

## 3. Results

The results of the base-case scenario, reflecting Croatian screening participation rates, are presented in Table 4. Both screening strategies, gFOBT and FIT, add life-years, decrease number of newly diagnosed patients, decrease deaths from colorectal cancer, and decrease costs compared to no screening option. Therefore, both screening options improve health outcomes and reduce overall costs, thus dominating no screening strategy (negative incremental costs per life-year gained, indicating savings per life-year gained).

In comparison to gFOBT, screening a cohort of 10,000 individuals aged 50 years with FIT is expected to prevent 2.5 additional cancer cases, decrease colorectal related cancer deaths by 0.9 deaths, and generate 2.2 additional life-years over the first five years after screening. Described health gains are accompanied by lower economic costs, indicating FIT dominating gFOBT as the preferred screening option.

Implementation of FIT to the national screening-eligible patients in Croatia (approximately 1,350,000 individuals; of whom 337,500 participate in screening), would, compared with no screening option, prevent 575 new colorectal cancer cases, reduce colorectal related cancer deaths by 255 and add 680 additional life-years over the first five years following screening.

The effectiveness of colorectal cancer screening is largely influenced by the screening participation rate and the uptake of follow-up colonoscopy after a positive test. Therefore, as shown in Table 5, two additional scenarios were analyzed: one in which screening participation rate was increased to 50%, and one in which the colonoscopy participation rate following a positive test was increased to 90%. The potential increase in both participation rates leads to additional health benefits and greater economic savings compared to the baseline scenario. This shows high future potential of a FIT colorectal cancer screening program in Croatia.

### Sensitivity Analyses

The developed model incorporates numerous inputs. Therefore, sensitivity analyses were performed to assess the robustness of the results (Figure 2). Results indicate high robustness of the model. The highest impact on projected costs per life-year gained was achieved by changing specificity, cost of colorectal cancer treatment, and sensitivity.

## 4. Discussion

This study estimated and analyzed the health outcomes and costs for the colorectal cancer screening in Croatia. The results demonstrate that gFOBT and FIT screening, compared with no screening, provide additional health benefits while reducing costs (negative incremental costs per life-year gained, i.e., savings per life-year gained). Among the screening options, FIT is expected to be the most cost-effective strategy. These study findings were supported by multiple sensitivity analyses, confirming the high robustness of the model.

Similar cost-effectiveness studies were conducted in several Central and Eastern European countries. In Austria, Jahn et al. compared the benefits and cost-effectiveness of four strategies (no screening, annual FIT for age 40–75, annual gFOBT for age 40–75, and 10-yearly colonoscopy for age 50–70) and concluded that an annual FIT or 10-year colonoscopy were the most effective options [28]. In Hungary, Csanádi et al. evaluated the implemented FIT colorectal cancer screening program and concluded that it has the potential to modestly reduce mortality rates [29]. In Slovakia, Babela et al. concluded that both biennial and annual FIT are cost-effective compared with no screening, leading to higher effectiveness at a reasonable increase in costs [30].

The conducted study has several limitations. As mentioned earlier, the retrieved data included a high proportion of patients with an unknown cancer stage. The allocation of this missing data to other cancer stages could affect treatment costs and model outcomes. Additionally, due to the unavailability of certain Croatia-specific data, inputs from international studies were used. Potential limitations of using foreign data include difficulties in comparability arising from differing clinical standards, healthcare systems, and levels of economic development. Where possible, data from meta-analyses were used to mitigate these biases.

An additional limitation is the relatively short five-year time horizon, which may not fully capture long-term health outcomes and associated costs, potentially affecting the accuracy of the cost-effectiveness estimates. Certain costs were also excluded from the model. Such costs include IT support for the National Colorectal Cancer Screening Program and certain intangible costs, for example, the psychological impact of colostomy. Mentioned inputs could not be adequately estimated due to a lack of reliable national or international data.

Last but not least, a successful colorectal screening program is a key prerequisite for better health outcomes. Screening also has a possibility of a significant cost reduction through increased detection of cancers in localized stages. For example, following introduction of screening programs, the Netherlands increased the proportion of localized cancer stage from 17% to 48%, while Slovenia achieved 49% localized cases with cancer screening program. Overall, higher screening and colonoscopy participation rates enhance program efficiency. In Croatia, this is recognized by the National Colorectal Cancer Screening Program, which aims to achieve a 60% participation rate in the target population and decrease mortality by 25% by 2030 [31].

The worldwide burden of colorectal cancer is expected to increase to 3.2 million new cases (a 63% increase) and 1.6 million deaths (a 73% increase) by 2040 [30]. Successful implementation and efficiency of screening strategies could prevent additional new cancer cases and deaths. In Croatia, the introduction of FIT screening option has the potential to increase screening participation rates, improve health outcomes and decrease related healthcare costs and economic burden associated with colorectal cancer.

## 5. Conclusions

In the Republic of Croatia, the National Colorectal Cancer Screening Program targets individuals aged 50–74 years using biennial screening with the gFOBT test. The aim of this study is to analyze the costs associated with colorectal cancer treatment and screening and to compare three screening scenarios in Croatia: no screening, biennial gFOBT test for individuals aged 50–74, and biennial FIT test for individuals aged 50–74 years. Calculation of annual per patient colorectal cancer costs indicates a substantial economic burden during first two years following diagnosis, with significantly higher costs observed in patients with advanced disease stage.

Model results indicate that FIT test could be the most cost-effective screening option and findings are supported by several sensitivity analyses. Successful implementation of a FIT based colorectal screening program could therefore increase screening participation and reduce both the clinical and economic burden of colorectal cancer in Croatia.

## Figures and Tables

**Figure 1 cancers-18-00150-f001:**
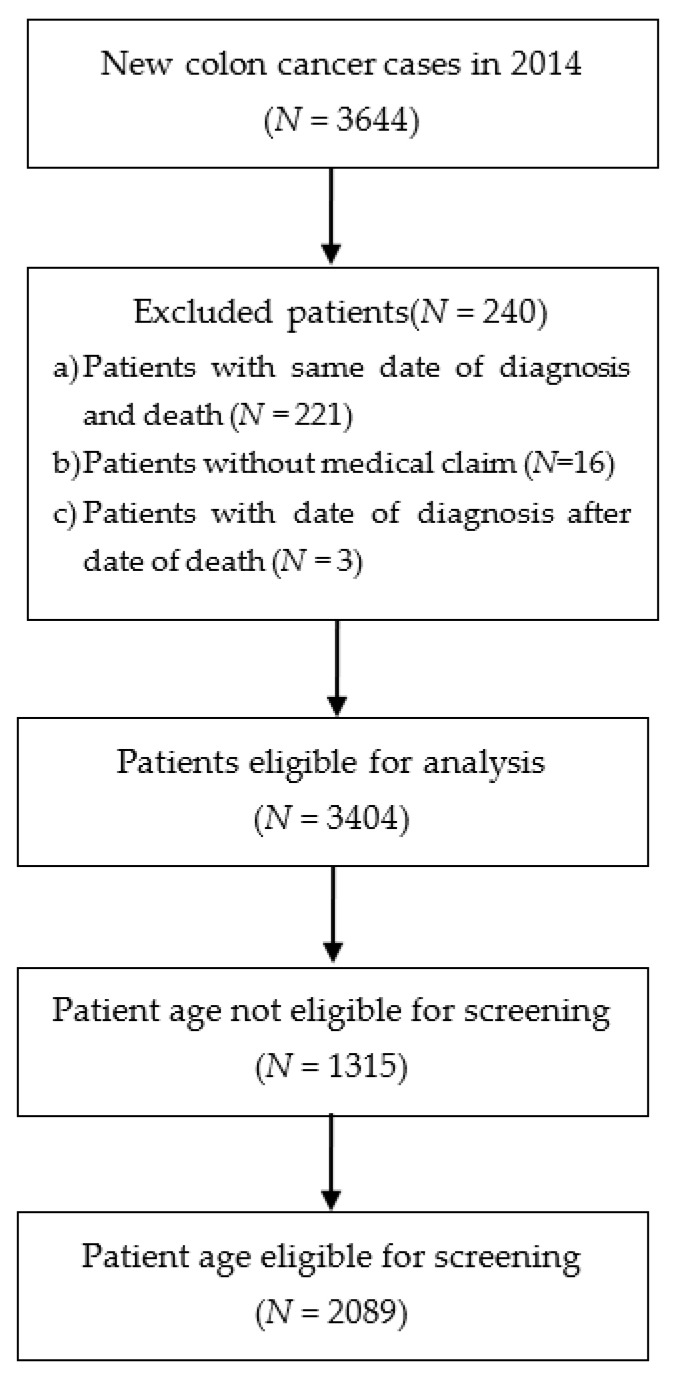
Model flow diagram.

**Figure 2 cancers-18-00150-f002:**
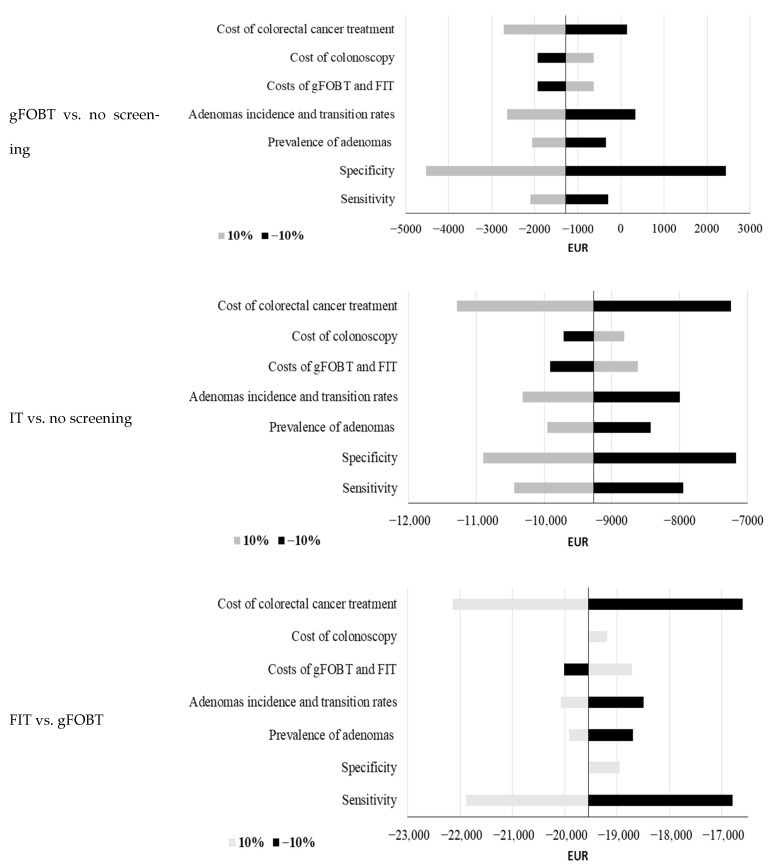
One-way sensitivity analysis of costs (savings) per life-year gained for basic scenario gFOBT-guaiac fecal occult blood test, FIT-fecal immunochemical test. Specificity is varied less than +10%, since it would be higher than 100% (it is limited to 100% for +10% scenario).

**Table 1 cancers-18-00150-t001:** Costs for colorectal cancer patients five year since diagnosis (total costs, EUR).

Healthcare Type	Costs in Euros Incurred in Each Year After Diagnosis					% of Total
	Year 1	Year 2	Year 3	Year 4	Year 5	
Inpatient care	14,101,997	4,979,774	1,744,014	1,043,426	815,091	50.8
Outpatient care *	3,673,798	2,485,502	1,387,713	1,022,718	859,201	21.1
Orthopedic and other medical prostheses	3,543,874	2,007,942	1,434,676	1,098,241	920,839	20.2
Prescription drugs	1,612,471	379,485	247,178	185,239	147,018	5.8
Primary care **	245,759	84,873	50,052	42,818	44,318	1.0
Sick leave ***	272,669	98,685	65,082	26,946	10,236	1.1
Total ****	23,450,568	10,036,261	4,928,715	3,419,388	2,796,703	
% of total	52.5	22.5	11.0	7.7	6.3	100.0

* Outpatient specialist care does not include primary healthcare. ** Certain services are paid using capitation system and cannot be attributed to the individual patient. Primary care includes general practitioners, laboratories, home nursing. *** Sick leaves paid by the Croatian Health Insurance Fund. **** Total costs in 5 years amounted to EUR 44,631,635.

**Table 2 cancers-18-00150-t002:** Costs of colorectal cancer per patient five years since diagnosis (EUR).

All patients						
Disease stage	Costs in euros during year after diagnosis (EUR)					Five years total
	Year 1	Year 2	Year 3	Year 4	Year 5	
Unknown	6065	3822	2029	1991	1670	15,576
Localized	4801	2281	1565	1165	1095	10,907
Regional	6786	3820	2303	1974	1850	16,732
Distant	11,460	11,695	7003	5121	4522	39,802
Total average	6889	4091	2337	1883	1697	16,897
Patients at age eligible for screening (age 50–74)						
Disease stage	Costs in euros during year after diagnosis (EUR)					Five years total
	Year 1	Year 2	Year 3	Year 4	Year 5	
Unknown	7029	4244	2019	1976	1651	16,919
Localized	5314	2474	1729	1207	1130	11,854
Regional	7786	4501	2781	2427	2335	19,830
Distant	14,086	13,021	8086	5607	5541	46,341
Total average	8003	4706	2702	2146	1975	19,533

**Table 3 cancers-18-00150-t003:** Inputs of the model.

Variable	Value	Source
**Screening participation (%)**		
Screening participation rates	25	National Screening Program
Colonoscopy after positive test result	75	National Screening Program
**gFOBT (%)**		
Cancer sensitivity	69.0	[20]
Adenoma sensitivity	9.5	[21]
Advanced adenoma sensitivity	14.2	[22]
Specificity	92.0	[22]
**FIT (%)**		
Cancer sensitivity	87.2	[22]
Adenoma sensitivity	7.6	[23]
Advanced adenoma sensitivity	36.7	[22]
Specificity	92.8	[22]
**Prevalence and incidence (%)**		
Adenoma prevalence	17.5	[24]
Advanced adenoma prevalence	8.4	[24]
Adenoma incidence	2.0	[25]
Adenoma to advanced adenoma transition rate	3.6	[26]
Advanced adenoma to cancer transition rate	1.8	[26]
**Test costs (EUR)**		
gFOBT	7.33	* Author′s calculation
FIT	13.09	* Author′s calculation
Colonoscopy without polypectomy	65.61	Published prices of the CHIF [27]
Colonoscopy with polypectomy	225.42	Published prices of the CHIF [27]

gFOBT-guaiac fecal occult blood test, FIT-Fecal immunochemical test. * Author′s calculation based on licitated test prices and other test related costs.

**Table 4 cancers-18-00150-t004:** Projection of health outcomes and costs for 10,000 patients 5 years after screening introduction.

Outcome	No screening	gFOBT	FIT
New colorectal cancer cases	85.7	84.0	81.5
Deaths from colorectal cancer	29.8	28.9	27.9
Life-years gained compared to no screening	-	2.8	5.0
Costs (EUR)	1,081,839	1,078,325	1,034,812
	gFOBT vs. no screening	FIT vs. no screening	FIT vs. gFOBT
Costs per life-year gained in scenario EUR	−1246	−9322	−19,548

gFOBT-guaiac fecal occult blood test, FIT-fecal immunochemical test.

**Table 5 cancers-18-00150-t005:** Projection of health outcomes and costs for 10,000 patients 5 years after screening introduction with higher participation rates.

Screening participation rate: 50%, colonoscopy participation: 75%			
**Outcome**	**No screening**	**gFOBT**	**FIT**
New colorectal cancer cases	85.7	82.2	77.3
Deaths from colorectal cancer	29.8	27.9	26.1
Life-years gained compared to no screening	-	5.6	10.1
Costs (EUR)	1,081,839	1,075,575	988,644
Screening participation rate: 25%, colonoscopy participation: 90%			
**Outcome**	**No screening**	**gFOBT**	**FIT**
New colorectal cancer cases	85.7	83.6	80.7
Deaths from colorectal cancer	29.8	28.7	27.6
Life-years gained compared to no screening	-	3.4	6.0
Costs (EUR)	1,081,839	1,074,406	1,019,329

gFOBT-guaiac fecal occult blood test, FIT-Fecal immunochemical test.

## Data Availability

The original contributions presented in this study are included in the article. Further inquiries can be directed to the corresponding author(s).

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
