# Peer review of "The Cost-Effectiveness of Organized National Colorectal Cancer Screening Program in Croatia"

_cancers, 2025, doi:10.3390/cancers18010150_

Round 1
Reviewer 1 Report (Previous Reviewer 2)
Comments and Suggestions for Authors
The authors reconsidered the entire manuscript and considerably improved it from the previous version.
I find it suitable for publication
Comments on the Quality of English Languagenone
Author Response
Comments 1: The authors reconsidered the entire manuscript and considerably improved it from the previous version.
I find it suitable for publication
Response 1: We thank to reviewer 1 on statement that he/she find now new version of manuscript suitable for publication due to improvement it from the previous version.
Reviewer 2 Report (New Reviewer)
Comments and Suggestions for Authors
The authors of this study, titled "The cost-effectiveness of organized national colorectal cancer screening program in Croatia," have attempted to analyze the costs of colorectal screening in Croatia and compare three screening scenarios to evaluate their cost-effectiveness for people aged 50 to 74 years. Although this study is of high interest to readers, there are issues that need to be addressed to make this suitable for publication in Cancers. Please find my comments below:
- The introduction section could include a brief description of both gFOBT and FIT tests, explaining the differences and their importance in colorectal cancer screening, which would make the introduction more informative and interesting for the readers.
- The study analysis included 3404 patients, but 1/3 of them (about 1300 patients) were not of an eligible age for screening. The authors failed to explain why such a large number of patients who were less than 50-74 years of age were included in the study, while the main aim and the focus of the study is to compare screening scenarios for people aged 50-74 years.
- Also, the authors could add a figure to describe the inclusion and exclusion criteria clearly, if possible.
- In Table 4 and in the Results section, lines 271-273, it is not clear whether the authors are discussing a decrease in new cancer cases by 2.5% or just 2.5?? Such confusions, if present elsewhere in the manuscript must be corrected.
- Figure 1, middle panel scale is not very clear as it presents only negative numbers compared to the top panel which presents both positive and negative numbers. The current quality of the figure is poor, and therefore, a high-resolution image is required to understand the data.
- The major weakness of this study (in addition to others the authors have listed in the discussion section), in my opinion, is the relatively short time horizon of 5 years for the projection of health outcomes and costs, as it would be crucial to have an understanding of the health outcome and cost of at least 10 or more years after the screening age of 50 years.
- Overall, the manuscript has merits but suffers from all the issue mentioned above. Addressing these issues would make the manuscript suitable for publication in Cancers.
In terms of the English language, the manuscript has many grammatical errors, spelling mistakes, and poorly structured sentences, all of which make the reading of this manuscript very hard. These errors must be corrected. For example, in line 69, "on the used screening test", line 70, "due to many available screening tests", line 86, "has the inclusion of FIT test potential to be cost-effective", and so on. These are just a few examples, and the manuscript has many more such issues that must be corrected.
Author Response
Reviewer 2
The authors of this study, titled "The cost-effectiveness of organized national colorectal cancer screening program in Croatia," have attempted to analyse the costs of colorectal screening in Croatia and compare three screening scenarios to evaluate their cost-effectiveness for people aged 50 to 74 years. Although this study is of high interest to readers, there are issues that need to be addressed to make this suitable for publication in Cancers. Please find my comments below:
Comment 1: The introduction section could include a brief description of both gFOBT and FIT tests, explaining the differences and their importance in colorectal cancer screening, which would make the introduction more informative and interesting for the readers.
Response 1: Brief description of gFOBT and FIT is added to the introduction.
Comment 2: The study analysis included 3404 patients, but 1/3 of them (about 1300 patients) were not of an eligible age for screening. The authors failed to explain why such a large number of patients who were less than 50-74 years of age were included in the study, while the main aim and the focus of the study is to compare screening scenarios for people aged 50-74 years.
Response 2: In the study, costs of colorectal cancer are described for all patients and separately for patients eligible for screening (50-74 years). Model for cost-effectiveness include only costs for patients aged 50-74 years. This is described in table 2 and supported by text.
Because the screening and model focuses on individuals aged 50–74 years, cost estimates specific to this age group were applied in the model (sentence in the 2.6. part of the article).
Comment 3: Also, the authors could add a figure to describe the inclusion and exclusion criteria clearly, if possible.
Response 3: Figure 1 is added to text.
Comment 4: In Table 4 and in the Results section, lines 271-273, it is not clear whether the authors are discussing a decrease in new cancer cases by 2.5% or just 2.5?? Such confusions, if present elsewhere in the manuscript must be corrected.
Response 4: The number refers to cancer cases, therefor it did not have a % sign. We rewrote such sentences, hopefully it is clearer now.
Comment5: Figure 1, middle panel scale is not very clear as it presents only negative numbers compared to the top panel which presents both positive and negative numbers. The current quality of the figure is poor, and therefore, a high-resolution image is required to understand the data.
Response 5: The figures are now in higher resolution and should be clearer.
Comment 6: The major weakness of this study (in addition to others the authors have listed in the discussion section), in my opinion, is the relatively short time horizon of 5 years for the projection of health outcomes and costs, as it would be crucial to have an understanding of the health outcome and cost of at least 10 or more years after the screening age of 50 years.
Response 6: We are aware of this issue and addressed it in the discussion section:
Additional limitation is the relative short five-year time horizon, which may not fully capture long-term health outcomes and associated costs, potentially affecting the accuracy of the cost-effectiveness estimates.
Comment 7: Overall, the manuscript has merits but suffers from all the issue mentioned above. Addressing these issues would make the manuscript suitable for publication in Cancers.
Response 7: We hope the issues are more clarified in the revised manuscript.
Comment 8: In terms of the English language, the manuscript has many grammatical errors, spelling mistakes, and poorly structured sentences, all of which make the reading of this manuscript very hard. These errors must be corrected. For example, in line 69, "on the used screening test", line 70, "due to many available screening tests", line 86, "has the inclusion of FIT test potential to be cost-effective", and so on. These are just a few examples, and the manuscript has many more such issues that must be corrected.
Response 8: The article was reread. The mistakes were corrected, as well as many words and sentences. Hopefully reading is easier in updated version.
This manuscript is a resubmission of an earlier submission. The following is a list of the peer review reports and author responses from that submission.
Round 1
Reviewer 1 Report
Comments and Suggestions for Authors
The article studies the intervention costs of a screening program versus no intervention.
The article is very well contextualized.
The authors do not mention intangible costs, such as the psychological effects of a colostomy, etc. These costs should be evaluated and analyzed.
It is important to note that adenomas are included; they are very common, and many of them do not progress to cancer or do so over a very long period of time. Furthermore, the fecal occult blood test has very low sensitivity, as the authors point out, and this can bias the results. In situ and intramucosal carcinomas are very often found in polyps and are treated only with polypectomy, which can lower costs and lead to bias. These histological subtypes should be separated from early cancer because they are not true cancers. These types of non-invasive carcinomas, by definition, are non-metastatic, and additional imaging tests are not required. There's also no mention of whether stage II carcinomas meet the criteria for adjuvant treatment. I think it would be appropriate to subdivide stage II carcinomas into these two groups and analyze the cost of each. The failure to consider the transition from one stage to another is a major limiting factor because it can involve quite significant costs. Keep in mind that colon cancer is a progressive disease, and it is easier for a regional cancer to progress to an advanced stage than for a localized cancer to progress to a regional or advanced stage, especially if adenomas within the localized cancer are taken into account. I believe the cost analysis is invalid. A prospective study could improve this analysis.
Author Response
Comments 1: The article studies the intervention costs of a screening program versus no intervention. The article is very well contextualized.
The authors do not mention intangible costs, such as the psychological effects of a colostomy, etc. These costs should be evaluated and analyzed.
Response 1: The potential intangible costs are added to the discussion. They are not added to the model since they do not have a direct cost and there is a lack of data for this kind of costs.
Comments 2: It is important to note that adenomas are included; they are very common, and many of them do not progress to cancer or do so over a very long period of time.
Response 2: The article analyses national colorectal cancer screening program. As a consequence of adhering to the programme, in certain patients adenomas are found through colonoscopy. As mentioned, very small number of adenomas progress to cancer, but certain small percentage does progress. Therefor, the effect of adenomas is twofold: a) these patients have a higher costs of colonoscopy than patients without adenomas since colonoscopy is paid at a higher cost if adenomas are removed during colonoscopy procedure (Croatian price system) and b) due to the adenoma removals, number of potential patients with cancer decreases, leading to potential lower cost of treatment cancer patients than not having a national screening programme.
Comments 3: Furthermore, the fecal occult blood test has very low sensitivity, as the authors point out, and this can bias the results.
Response 3: The sensitivity and specificity of each test are based on a literature review and published articles. Since sensitivity and specificity influence results, they are varied in the sensitivity analyses.
Comments 4: In situ and intramucosal carcinomas are very often found in polyps and are treated only with polypectomy, which can lower costs and lead to bias. These histological subtypes should be separated from early cancer because they are not true cancers. These types of non-invasive carcinomas, by definition, are non-metastatic, and additional imaging tests are not required.
Response 4: The model is based on following diagnoses: C18 (malignant neoplasm of colon), C19 (malignant neoplasm of rectosigmoid junction), C20 (malignant neoplasm of rectum) and C21 (malignant neoplasm of anus and anal canal). The in-situ patients have a D01 diagnosis in the Croatian National Cancer Registry and their data (costs) were not taken into model.
Comments 5: There's also no mention of whether stage II carcinomas meet the criteria for adjuvant treatment. I think it would be appropriate to subdivide stage II carcinomas into these two groups and analyze the cost of each.
Response 5: The data about treatment criteria (if patients meet the criteria for adjuvant treatment) are not collected by the The Croatian National Cancer Registry. These data could be potentially retrieved from hospitals, but would be based on personal patient data and their medical records. Due to the sensitivity of these personal data and data restrictions (GDPR in European Union), this sensitive data would be almost impossible to retrieve. Therefore, only anonymous patient data were used in the model, without access to medical records.
Comments 6: The failure to consider the transition from one stage to another is a major limiting factor because it can involve quite significant costs. Keep in mind that colon cancer is a progressive disease, and it is easier for a regional cancer to progress to an advanced stage than for a localized cancer to progress to a regional or advanced stage, especially if adenomas within the localized cancer are taken into account. I believe the cost analysis is invalid. A prospective study could improve this analysis.
Response 6: We acknowledge this disadvantage and have mentioned it in the paper. Currently, The Croatian National Cancer Registry gathers data at a time of diagnosis and has only this diagnose (does not collect data regarding disease progression). Disease progression could potentially be accessed through hospital medical records, but, as mentioned before, access to the data would be very challenging due to the collection of personal data.
Additionally, the retrieved costs are based on initial diagnosis and, based on retrieved and possible data without medical records, could not be separated on the basis of disease progression. We opted not to use international data since we could not calculate costs for and without disease progression due to the lack of data. Additionally, each country has a different percentage of patients in certain cancer stages (Croatia having a higher percentage of patients discovered with advanced cancer stage), therefor leading to the potential data bias if international data is used.
Reviewer 2 Report
Comments and Suggestions for Authors
The article is well written and offer a projection over the Croatian population on how CRC screening will impact the economic burden. As mentioned in the introduction section, similar scenarios have been taken into account before. The question is, what does your article bring to the table that the other EU countries do not? What are the major points of the article since it already has been demonstrated that FIT will reduce costs.
Also:
What is the cut-off value used ofr gFOBT and FIT?
Is there any IT/registry of the available screening population?
Are there any internal checks and external validation to local registry data?
Did you take into account the idea of repeating the FIT in patients? Would this increase costs?
Author Response
Comments 1: The article is well written and offer a projection over the Croatian population on how CRC screening will impact the economic burden. As mentioned in the introduction section, similar scenarios have been taken into account before. The question is, what does your article bring to the table that the other EU countries do not? What are the major points of the article since it already has been demonstrated that FIT will reduce costs.
Response 1: According to the conducted literature review, FIT is cost-effective in most models. In some cases it is also a cheaper option, while in some cases is more expensive, but falls within cost-effective boundary of a relevant country.
Croatia is currently one of the worst EU countries regarding incidence and mortality of colorectal cancer. The introduction of FIT test, a as more quality screening method than current gFOBT, has a potential to be cost-efficient and significantly improve treatment outcomes of diagnosed patients. Our conducted literature review found lacking data and studies regarding Eastern and Southern Europe, as well as countries that entered European union after 2003. Beside cost-effectiveness analysis, we have also conducted thorough analysis of costs for colorectal cancer patients in Croatia, using best possible data source (claims and data from the only national compulsory insurance organization).
Comments 2:
What is the cut-off value used of gFOBT and FIT?
Response 2: According to the current practise and guaic-based method, gFOBT has no cutt-offs, just positive or negative.
Comments 3: Is there any IT/registry of the available screening population?
Response 3: Yes, there is a screening IT registry (the National Colon Cancer Screening Program). It differs from the The Croatian National Cancer Registry which collects data about newly diagnosed cancer patients.
Comments 4: Are there any internal checks and external validation to local registry data?
Response 4: The are certain validations, but registry data mostly depends on the retrieved data from different sources described in the article. Certain discovered data errors (missing data, incomplete data) are analysed and solved by retrieving additional patient data. Due to the unavailability of national medical records in the electronic form, most data limitations cannot be corrected without accessing hospital data. Patients with obvious data limitations were excluded from the model, as described in the article.
Comments 5: Did you take into account the idea of repeating the FIT in patients? Would this increase costs?
Response 5: During model planning, we have discussed potential number of tests. Since current screening program gFOBT is based on one test (one price per patient) and this will probably be the case with FIT implementation, we have opted for one FIT test. The possible effect of price change is tested in the sensitivity analyses. The add of additional FIT would increase the costs of testing itself since FIT is more expensive than gFOBT, but could also “catch” additional cancer patients.
Reviewer 3 Report
Comments and Suggestions for Authors
Dear Authors,
The manuscript addresses an important public health and health economics topic,the cost-effectiveness of colorectal cancer screening in Croatia. But there are areas that could be improved.
The introduction is informative but overly long and at times reads like a review article. It should be more concise, with a sharper focus on the specific knowledge gap in Croatia. The study aim is clear, but the research question and hypothesis should be articulated more directly.
The methodology is described in detail, yet a few aspects require clarification. The decision model is based on 10,000 simulated individuals over only five years, which may be too short to capture the full benefits of CRC screening given the typically slower progression. Several transition probabilities and test characteristics are borrowed from non-Croatian sources.While this is acknowledged, the potential bias should be discussed more explicitly. The costing approach is robust, using CHIF data, but the rationale for applying a 3% discount rate should be justified (or sensitivity-tested against other rates). Handling of missing data (patients with unknown stage) also deserves further explanation.
The results are presented clearly but could be synthesized more effectively. For example, highlighting policy-relevant numbers upfront (such as cost per life-year gained in relation to Croatia’s GDP per capita) would make the findings more impactful for decision-makers.
The discussion sometimes repeats results rather than engaging critically with them. I recommend tightening the narrative, avoiding repetition, and strengthening the critique of limitations. The implications of using foreign parameters and of the limited time horizon are underexplored. The comparison with other countries is useful but could be structured more systematically, with emphasis on maybe regional comparators.
The conclusions are in line with the findings but somewhat overstated, given the limitations. While FIT is likely more cost-effective, the evidence is still model-based and relies partly on external inputs. The conclusions should be reframed more cautiously, emphasizing that the results support a shift toward FIT, while further real-world data and longer-term evaluation are still needed.
There are some inconsistencies in formatting, as well as minor grammar mistakes. Abbreviations are numerous and should be reduced or used consistently.
Comments on the Quality of English LanguageI am not a native English speaker and therefore not fully competent to judge the language in detail. However, I noticed some inconsistencies and minor grammatical issues throughout the manuscript.
Author Response
Comments 1: The manuscript addresses an important public health and health economics topic, the cost-effectiveness of colorectal cancer screening in Croatia. But there are areas that could be improved. The introduction is informative but overly long and at times reads like a review article. It should be more concise, with a sharper focus on the specific knowledge gap in Croatia.
Response 1: The introduction is shortened and should be more concise.
Comments 2: The study aim is clear, but the research question and hypothesis should be articulated more directly.
Response 2: The research question and hypothesis are revised.
Comments 3: The methodology is described in detail, yet a few aspects require clarification. The decision model is based on 10,000 simulated individuals over only five years, which may be too short to capture the full benefits of CRC screening given the typically slower progression.
Response 3: We understand this issue and have additionally mentioned it in the paper discussion. As mentioned below, we had to user certain non-Croatian data, and with additional years this would be even challenging.
Comments 4: Several transition probabilities and test characteristics are borrowed from non-Croatian sources. While this is acknowledged, the potential bias should be discussed more explicitly.
Response 4: We have discussed this more explicitly in the revised version.
Comments 5: The costing approach is robust, using CHIF data, but the rationale for applying a 3% discount rate should be justified (or sensitivity-tested against other rates).
Response 5: We have added literature sources for the usage of this discount rate.
Comments 6: Handling of missing data (patients with unknown stage) also deserves further explanation.
Response 6: To clarify this matter additionally, we have added following sentences at the end of part 2.4. (Population):
The exact stage of disease for patients with unknown stage would require access to the individual patient data and medical records. Since this data would be hardly accessible due to the data sensitivity and General Data Protection Regulation in the European Union, the unknown stage is used as a separate (additional) disease stage in the analysis of costs and cost-effectiveness.
Also, in part 2.6. (Model and Inputs):
The relative survival probabilities of unknown stage were calculated as an average for localized, regional and distant cancer stage probabilities.
Comments 7: The results are presented clearly but could be synthesized more effectively. For example, highlighting policy-relevant numbers upfront (such as cost per life-year gained in relation to Croatia’s GDP per capita) would make the findings more impactful for decision-makers.
Response 7: Since we do not have costs but savings per life-year gained, we presented the results only in monetary terms. Also, we have avoided using negative costs per year-life gained in the abstract since negative values could indicate higher costs and lower health related outcomes of new treatment. Our results are opposite (negative value implicates lower costs and higher health related outcomes), but since they are not as straightforward we have only used monetary values.
Comments 8: The discussion sometimes repeats results rather than engaging critically with them. I recommend tightening the narrative, avoiding repetition, and strengthening the critique of limitations. The implications of using foreign parameters and of the limited time horizon are underexplored. The comparison with other countries is useful but could be structured more systematically, with emphasis on maybe regional comparators.
Response 8: The discussion has been revised based on suggestions.
Comments 9: The conclusions are in line with the findings but somewhat overstated, given the limitations. While FIT is likely more cost-effective, the evidence is still model-based and relies partly on external inputs. The conclusions should be reframed more cautiously, emphasizing that the results support a shift toward FIT, while further real-world data and longer-term evaluation are still needed.
Response 9: The conclusion has been revised based on suggestions.
Comments 10: There are some inconsistencies in formatting, as well as minor grammar mistakes. Abbreviations are numerous and should be reduced or used consistently.
Response 10: The article is re-read and hopefully inconsistencies and errors are minimised. The abbreviations are reduced and mostly include gFOBT and FIT.